# A comparison of OH nightglow volume emission rates as measured by SCIAMACHY and SABER

Yajun Zhu[1, 2], Martin Kaufmann[1, 3], Qiuyu Chen[1, 3], Jiyao Xu[2, 4], Qiucheng Gong[1, 3], Jilin Liu[1, 5], Daikang Wei[1, 3], and Martin Riese[1, 3]

[1]Institute of Energy and Climate Research, Forschungszentrum Jülich, Jülich, Germany
[2]State Key Laboratory of Space Weather, National Space Science Center, Chinese Academy of Sciences, Beijing, China
[3]Institute for Atmospheric and Environmental Research, University of Wuppertal, Germany
[4]School of Astronomy and Space Science, University of Chinese Academy of Sciences, Beijing, China
[5]Qian Xuesen Laboratory of Space Technology, China Academy of Space Technology, Beijing, China

**Correspondence:** Qiuyu Chen (q.chen@fz-juelich.de)

**Abstract.** Hydroxyl (OH) short-wave infrared emissions arising from OH(4-2, 5-2, 8-5, 9-6) as measured by channel 6 of the SCanning Imaging Absorption spectroMeter for Atmospheric CHartographY (SCIAMACHY) are used to derive concentrations of OH(v=4, 5, 8, and 9) between 80 km and 96 km. Retrieved concentrations are used to simulate OH(5-3, 4-2) integrated radiances at 1.6 μm and OH(9-7, 8-6) at 2.0 μm as measured by the Sounding of the Atmosphere using Broadband Emission Radiometry (SABER) instrument, which are not fully covered by the spectral range of SCIAMACHY measurements. On average, SABER "unfiltered" data is on the order of 40% at 1.6 μm and 20% at 2.0 μm larger than the simulations using SCIAMACHY data. "Unfiltered" SABER data is a product, which accounts for the shape, width, and transmission of the instrument's broadband filters, which do not cover the full ro-vibrational bands of the corresponding OH transitions. It is found that the discrepancy between SCIAMACHY and SABER data can be reduced by up to 50%, if the filtering process is carried out manually using published SABER interference filter characteristics and latest Einstein coefficients from the HITRAN database. Remaining differences are discussed with regard to model parameter uncertainties and radiometric calibration.

*Copyright statement.* TEXT

## 1  Introduction

Hydroxyl (OH) airglow stems from spontaneous emissions of metastable excited OH molecules which are mainly produced by the exothermic reaction of H and $O_3$ in the upper mesosphere and lower thermosphere (UMLT). Its emission layer peaks at an altitude of approx. 87 km and extends about 8 km (Baker and Stair, 1988; Hong et al., 2010). OH airglow covers a broad spectral region from the ultraviolet to near-infrared spectral range and is of importance for studying photochemistry and dynamics in the UMLT region.

Since first confirmed by Meinel (1950), OH airglow emissions have been widely observed using various remote spectro-scopic techniques (e.g., Offermann and Gerndt, 1990; von Savigny et al., 2004; Kaufmann et al., 2008; Smith et al., 2010; Zhu et al., 2012). The measurements obtained in such studies have been analyzed for various purposes. For example, rotational temperature can be obtained from OH emissions as a proxy for kinetic temperature under the assumption of rotational local thermodynamic equilibrium (LTE) (Offermann et al., 2010; Zhu et al., 2012; Liu et al., 2015). Gravity waves passing through the OH airglow layer can be monitored to study the dynamics and energy balance in the UMLT (Xu et al., 2015). The under-standing of OH relaxation mechanisms with different species can be improved by studying different OH band emissions in the UMLT (Kaufmann et al., 2008; Xu et al., 2012; von Savigny et al., 2012). Another important application of OH airglow is to derive trace constituents in the UMLT, such as H and O abundances (Kaufmann et al., 2013; Mlynczak et al., 2018; Panka et al., 2018; Zhu and Kaufmann, 2018).

OH nightglow has been globally measured, among others, by SABER (Sounding of the Atmosphere using Broadband Emission Radiometry) operating since 2002, and SCIAMACHY (SCanning Imaging Absorption spectroMeter for Atmospheric CHartographY) observing from 2002 to 2012. SABER performed observations successfully over a 17-year period, covering one and half solar cycles, and is still measuring, and many outstanding achievements have been accomplished using these data (e.g., Xu et al., 2007; Smith et al., 2008; Mlynczak et al., 2010; Hong et al., 2010). The OH data obtained by SABER have been used by different investigators (Smith et al., 2010; Mlynczak et al., 2013; Panka et al., 2018; Mlynczak et al., 2018) to derive atomic oxygen abundance in the UMLT; however, deviations of up to 60% were found in comparison with atomic oxygen data derived from $O(^1S)$ green-line measurements obtained by SCIAMACHY and WINDII (Wind Imaging Interferometer) (Kaufmann et al., 2014; Zhu et al., 2015). This large deviation promoted a discussion on the absolute values of atomic oxygen abundance (Mlynczak et al., 2018; Panka et al., 2018; Zhu and Kaufmann, 2018). Mlynczak et al. (2018) derived new atomic oxygen data from SABER OH 2.0 $\mu$m absolute radiance measurements in the UMLT under the constraints of the global annual mean energy budget. Panka et al. (2018) also retrieved atomic oxygen data from SABER OH 1.6 $\mu$m and 2.0 $\mu$m radiance ratios as an alternative approach. Further new atomic oxygen data were recently derived by Zhu and Kaufmann (2018) from SCIAMACHY nighttime OH(9-6) band measurements using rate constants measured in the laboratory by Kalogerakis et al. (2016), which agree with atomic oxygen data derived from SCIAMACHY $O(^1S)$ green-line and $O_2$ A-band measurements within a range of 10-20% (Zhu and Kaufmann, 2019). While the agreement between new atomic oxygen data obtained by SABER and SCIAMACHY has improved, systematic deviations of up to 50% still persist (Zhu and Kaufmann, 2018). This systematic difference needs to be addressed in future studies.

In this study, OH nightglow limb spectra measured by SCIAMACHY were used to derive OH spectrally averaged radi-ances at 1.6 $\mu$m and 2.0 $\mu$m as measured by SABER. The obtained radiances were compared to SABER OH radiometric measurements to investigate whether systematic differences exist between the two datasets.

## 2 OH nightglow measurements and auxiliary data

From 2002 to 2012, OH Meinel-band near-infrared emissions were measured simultaneously by SCIAMACHY on the Envisat and by SABER on the TIMED (Thermosphere, Ionosphere, Mesosphere Energetics and Dynamics) satellite. The spectral range of both instruments covers several OH emission bands stemming from different vibrational states (Kaufmann et al., 2008; Mlynczak et al., 2013). The SCIAMACHY instrument on Envisat operated in a sun-synchronous orbit with an equator crossing local solar time of 10 a.m./p.m. The limb spectra used here were observed by SCIAMACHY in a dedicated mesosphere/thermosphere mode and the limb observational range covered 24 tangent altitudes from 73 km to 148 km with a vertical sampling of 3.3 km. SCIAMACHY was a multi-channel grating spectrometer and its channel 6 measured OH spectra arising from upper vibrational states in the range of 2 to 9 at a spectral resolution of 1.5 nm. The measurement error of SCIAMACHY channel 6 is about 1.2% (Zoutman et al., 2000). Channel 6 covers a spectral range from 971 nm to 1773 nm (Lichtenberg et al., 2006). In this study, only the spectral range of channel 6 up to 1589 nm was used due to the reduced performance of the detector beyond this wavelength (Lichtenberg et al., 2006). It should be noted that SCIAMACHY channel 7 and 8 covered spectral ranges of 1934-2044 nm and 2259-2386 nm, respectively, but unfortunately suffered from ice condensation on their detectors (Lichtenberg et al., 2006).

SABER is a multi-channel radiometer and observes radiometric OH(9-7, 8-6) ro-vibrational lines with wavelengths around 2.0 μm (channel 8) and OH(5-3, 4-2) band emissions at about 1.6 μm (channel 9) (Xu et al., 2012). The altitude range of the observation covers 60-180 km with vertical resolution of approx. 2 km (Mertens et al., 2009). To the authors' knowledge, there are no publicly available references on the observed accuracy of the SABER OH channels, except for a presentation named "SABER Instrument Performance and Measurement Requirements" published on http://saber.gats-inc.com/overview.php, which is the official source of SABER data products. According to this document, the estimated accuracy of the 1.6 and 2.0 μm channel data is about 3% at 80-90 km and about 20% at 90-100 km. Since the SABER instrument is a radiometer, individual OH ro-vibrational emission lines cannot be resolved. Figure 1 shows simulated OH airglow emissions in the spectral range between 1000 nm and 2400 nm; spectral ranges covered by the instruments are shaded in different colors.

Since the spectral coverage of SCIAMACHY and SABER does not coincide, we can not compare their measurements directly. However, both instruments observed ro-vibrational lines stemming from the same upper vibrational states. This offered us an opportunity to calculate the number densities of the OH upper vibrational states and then simulate the same ro-vibrational emission bands for the purposes of comparison. In our study, OH limb spectra measured by SCIAMACHY at 1078-1100 nm, 1297-1325 nm, 1377-1404 nm, and 1575-1588 nm were used, as shown in Figure 2. The spectral ranges covered ro-vibrational lines in the OH(5-2), OH(8-5), OH(9-6) and OH(4-2) bands, respectively, with low rotational quantum numbers $N$ ($N \leq 3$) to reduce the potential uncertainty that can be introduced by over-populated high-$N$ rotational states (Cosby and Slanger, 2007; Noll et al., 2015; Oliva et al., 2015); details are discussed later. From these measurements, number densities of OH($v$=4, 5, 8, and 9) are obtained, which are used to simulate corresponding SABER measurements.

For comparison, SABER V2.0 data were used, including OH in-band and "unfiltered" 1.6 μm and 2.0 μm data (Mlynczak et al., 2013). The in-band OH radiance data comprised raw data that did not take into account filter transmission, while the

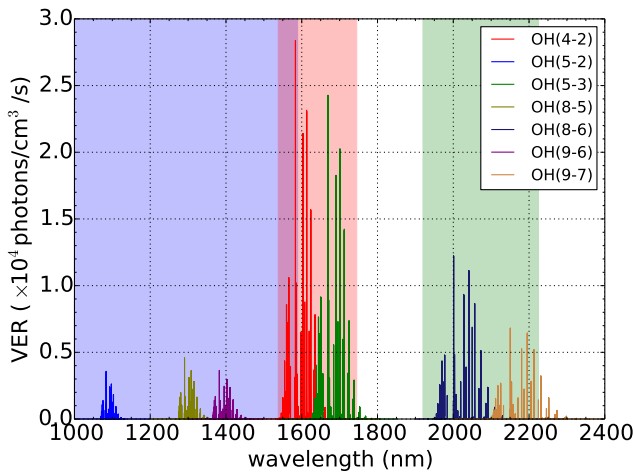

**Figure 1.** Simulated OH airglow emission bands used in this study in a spectral range between 1000 nm and 2400 nm. Shaded light blue region covers a spectral range observed by SCIAMACHY channel 6; shaded light red and light green regions cover two spectral ranges measured by SABER channel 9 and 8, respectively.

"unfiltered" OH radiance data consider the interference filter characteristics measured in the laboratory (Xu et al., 2012). The unfiltering process depends on the spectral shape of the underlying ro-vibrational distribution of the emission. This shape has to be determined by a model, which depends on the rotational temperature and on the transition probabilities (Einstein coefficients). Beside using the official in-band and "unfiltered" data, separate in-band and "unfiltered" datasets were obtained from the SCIAMACHY measurements, using the bandpass filter transmission of Baker et al. (2007) and various Einstein coefficient datasets, for details see later. In this procedure, we also considered OH(3-1) and OH(7-5) emission lines observed by SABER 1.6 μm and 2.0 μm channels, respectively, and their contributions to the two channels were calculated based on SCIAMACHY OH(3-1) and OH(7-4) measurements. In order to enhance the signal-to-noise ratio and to obtain a large number of coincident measurements with both instruments, monthly zonal median data in 5-degree latitude bins were used. Since the SCIAMACHY instrument can not measure nighttime temperature in the UMLT, co-located SABER measurements were also used here. The coincidence criteria selected were $\pm 2.5°$ in latitude and one hour in local time.

## 3   Methodology

### 3.1   OH emission model

The exothermic reaction of H and $O_3$ in the atmosphere was identified by Bates and Nicolet (1950) as the major source of vibrationally excited hydroxyl radicals ($OH^*$) near the mesopause region.

$$H + O_3 \rightarrow OH^*(v \leq 9) + O_2 \tag{R1}$$

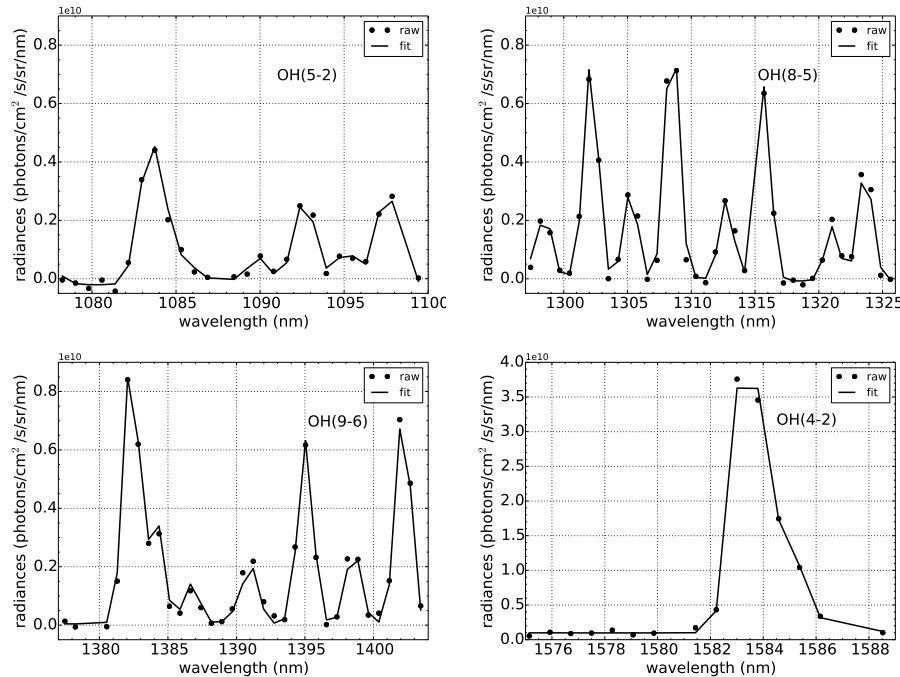

**Figure 2.** Monthly zonal median OH(4-2), OH(5-2), OH(8-5), and OH(9-6) limb spectra at tangent altitude about 86 km for September 2005 in a latitude range 35°N-40°N. raw: the raw limb spectra measured by SCIAMACHY; fit: simulated limb spectra as measured by SCIAMACHY from retrieval results.

Metastable excited OH* can be de-excited via radiative, chemical, and collisional relaxation processes. OH($v \leq 8$) is not only initially populated by the reaction of H + O₃, but is also produced by the deactivation of higher vibrational states of OH* via radiative relaxation and quenching. The number density of OH($v$) can be obtained from its emission line measurements by dividing by the corresponding Einstein coefficients. The volume emission rate $V_{v-v'}(i)$ of an arbitrary ro-vibrational line within a vibrational band OH($v - v'$) can be calculated as

$$V_{v-v'}(i) = n_v \cdot \frac{g_v(i) \cdot \mathrm{e}^{-E_v(i)/(k \cdot T)}}{Q_v(T)} \cdot A_{v-v'}(i) \tag{1}$$

$n_v$ is the total number density of the corresponding upper vibrational state $v$ and $A_{v-v'}(i)$ is the Einstein coefficient of the specific state-to-state transition from vibrational level $v$ to $v'$. $E_v(i)$ and $g_v(i)$ are the rotational energy and degeneracy of the upper rotational state of the $i$th line considered. k is the Boltzmann constant and T is the temperature. $Q_v(T)$ is the rotational partition sum of OH($v$). This formula is only valid under rotational local thermodynamic equilibrium (LTE) conditions and deviations are discussed later.

## 3.2 Retrieval model

SCIAMACHY measured integrated OH spectra along the line of sight in the tangent altitude range from approx. 73 km to approx. 149 km. The SCIAMACHY OH limb measurements can be expressed as

$$\boldsymbol{y} = \boldsymbol{F}(\boldsymbol{x}, \boldsymbol{b}) + \boldsymbol{\epsilon} \tag{2}$$

$\boldsymbol{y}$ corresponds to the measured SCIAMACHY OH limb spectra. $\boldsymbol{F}$ is the functional formula of the forward model involving equation 1. $\boldsymbol{x}$ represents the number densities ($n_v$) of the corresponding upper vibrational state of the emission lines. $\boldsymbol{b}$ is the parameter vector of the forward model, e.g., Einstein coefficients. $\boldsymbol{\epsilon}$ represents stochastic measurement errors. The retrieval can be regarded as an approach to solving an inversion problem in the presence of indirect measurements of the properties of interest. In our setup we assume that each atmospheric layer emits OH airglow homogeneously, and we set the retrieval grid to be identical to the tangent altitude grid of the averaged OH limb measurements. To improve the efficiency of the retrieval, to suppress noise in the solution, and to achieve a smooth transition of the retrieved quantities into model data at the upper boundary, a regularization term is added to the minimization (Rodgers, 2000):

$$\boldsymbol{x}_{i+1} = \boldsymbol{x}_a + (\boldsymbol{K}_i^T \boldsymbol{S}_\epsilon^{-1} \boldsymbol{K}_i + \boldsymbol{S}_a^{-1})^{-1} \boldsymbol{K}_i^T \boldsymbol{S}_\epsilon^{-1}[\boldsymbol{y} - \boldsymbol{F}(\boldsymbol{x}_i, \boldsymbol{b}) + \boldsymbol{K}_i(\boldsymbol{x}_i - \boldsymbol{x}_a)] \tag{3}$$

$\boldsymbol{x}_i$ reaches the optimal estimate solution when the retrieval converged. $\boldsymbol{K}_i$ corresponds to the first derivative matrix of the forward model, named the Jacobian matrix. $\boldsymbol{x}_a$ represents the a-priori knowledge of the total number densities of OH($v$), and $\boldsymbol{S}_a^{-1}$ is the regularization matrix. $\boldsymbol{S}_\epsilon$ is a diagonal error covariance matrix of $\boldsymbol{y}$.

# 4 Results and discussion

## 4.1 Error analysis

The confidence level of simulated volume emission rates (VERs) can be assessed by considering three main aspects: the uncertainty of the auxiliary atmospheric quantities, i.e., temperature; the uncertainty of rate constants, i.e., Einstein coefficients; and the potential uncertainty introduced by over-populated high rotational states due to non-local thermodynamic equilibrium (non-LTE) effects. The temperature uncertainty in the SABER measurements includes random and systematic errors; Dawkins et al. (2018) summarized SABER temperature uncertainties. We consider only systematic errors in SABER temperatures, because monthly mean data are used. The SABER systematic temperature uncertainty is approx. 1.5 K at 70-80 km, 4 K at 90 km, and 5 K at 100 km, respectively. Accordingly, VERs are affected by temperature uncertainties by less than 1% between 80 km and 96 km on average, as obtained by Xu et al. (2012) in their investigation of the temperature dependence of the band Einstein coefficients as well.

Many OH Einstein coefficient datasets can be found in the OH research community (see, e.g., Liu et al. (2015)). We consider the values given in the latest HITRAN molecular spectroscopic database (Gordon et al., 2017) and the OH Einstein A values calculated by van der Loo and Groenenboom (2007, 2008) and Brooke et al. (2016). The uncertainty of the Einstein coefficient

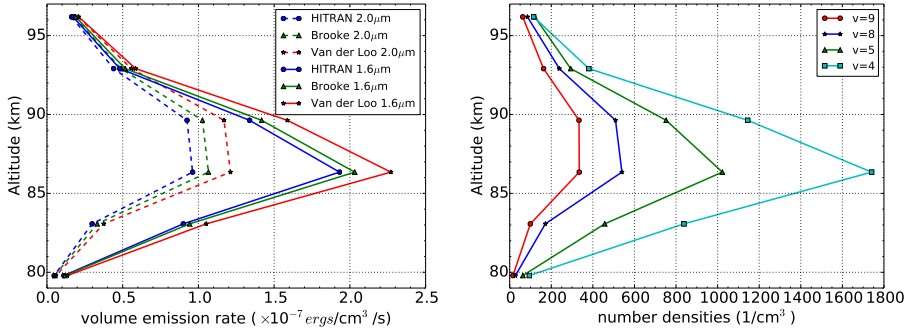

**Figure 3.** Simulated SABER "unfiltered" OH 1.6 μm and 2.0 μm volume emission rates from SCIAMACHY data using Einstein A values of HITRAN, Brooke et al. (2016), and van der Loo and Groenenboom (2007, 2008) at 20°N-40°N for October 2007 (left); Corresponding retrieved OH number densities of vibrational states 9, 8, 5, and 4 from SCIAMACHY data using HITRAN database (right).

affects simulated VERs in two ways: In the retrieval of vibrationally excited OH from SCIAMACHY data and in the simulation of the SABER measurements. Figure 3 (left) shows simulated SABER "unfiltered" OH 1.6 μm and 2.0 μm VER profiles obtained from SCIAMACHY measurements using these three Einstein coefficient datasets. Corresponding OH number density profiles as derived using Einstein coefficients from the HITRAN database at vibrational states 9, 8, 5, and 4 are also given.

Highest VERs are obtained by using the Einstein coefficients calculated by van der Loo and Groenenboom (2007, 2008) and the lowest are obtained by using the HITRAN database. The differences between them are approx. 26% for the simulation of SABER 2.0 μm VERs and approx. 19% for the simulation of SABER 1.6 μm VERs. Similar values were also obtained if we used data for other latitude bins or time periods. The same procedure was also applied to the simulation of SABER 2.0 μm and 1.6 μm in-band data, giving similar results.Therefore, we used these results as a proxy to estimate related uncertainties of the

Einstein coefficients.

Cosby and Slanger (2007); Noll et al. (2015); Oliva et al. (2015) reported that middle and higher excited rotational states ($N \geq 4$) of OH do not meet the LTE hypothesis and that these levels are overpopulated. An estimation of the non-LTE contribution was performed by Oliva et al. (2015) based on cross-dispersed cryogenic spectrometer measurements in the spectral range of 0.97 μm to 2.4 μm. A combination of two Boltzmann distribution equations with cold and hot OH rotational temperatures

was used to predict the observed intensities of OH emission lines. Kalogerakis et al. (2018) re-analyzed the data used by Oliva et al. (2015) to estimate the OH rotational temperatures following the approach taken by Cosby and Slanger (2007) and Oliva et al. (2015). They found that the thermalization of every OH vibrational level is incomplete.

The low spectral resolution of SCIAMACHY spectra does not allow to estimate this effect from the measured data. Therefore, we performed model simulations using the same approach and parameter sets as Oliva et al. (2015) to quantify the effect

of incomplete thermalization on the spectral ranges used in this study. We calculated OH 1.6 μm and 2.0 μm VERs by considering only the cold rotational temperature and then obtained them using cold and hot temperatures together as Oliva et al. (2015) and Kalogerakis et al. (2018) did. It was found that differences between them are less than 2% for both SABER channels. The SABER 1.6 μm and 2.0 μm channels also observe emission lines from OH(3-1) and OH(7-5), respectively. We estimated their

influence on spectrally integrated radiances by the derivation of the corresponding emissions using SCIAMACHY OH(3-1) and OH(7-4) nightglow measurements. These simulations show that the contributions of OH(7-5) and OH(3-1) to the two channels are about 3% and 1% on average, respectively.

In summary, the uncertainty of the Einstein coefficient dominates the error budget for the in-band and "unfiltered" data, which is on the order of 20% and 26% for the SABER 1.6 μm and 2.0 μm VER simulations, respectively.

## 4.2 Comparison of SABER measurements and simulations

An intercomparison between 1.6 μm and 2.0 μm in-band and "unfiltered" VERs as measured by SABER and corresponding simulations using SCIAMACHY data and HITRAN OH Einstein coefficients is given in Figure 4 for two different latitudes in September 2005. Error bars shown in Figure 4 represent the root mean square value of all uncertainties discussed in the subsection 4.1. The top two plots show a comparison of the "unfiltered" data and the bottom two figures show the in-band data. SABER measurements are always larger than the simulations using SCIAMACHY data. For the "unfiltered" data, deviations of SABER OH 1.6 μm measurements with respect to the corresponding simulations increase with altitude from 30-45% at 83 km to 55-80% at 96 km, depending on latitudes. The difference of SABER OH 2.0 μm measurements with respect to the corresponding simulations is 16% at 86 km. At 96 km, it reaches 70% in latitude bins 0°-20°N and approx. 90% in 20°N-40°N.

Surprisingly, for the in-band data, the differences for the 1.6 μm and 2.0 μm channels are significantly smaller at most altitudes. They vary in a range of 8-28% (21-50%) and 8-60% (28-100%) from 83 km to 96 km at 0°-20°N (20°N-40°N). It should be noted that SCIAMACHY and SABER have a resolution of about 3.3 km and 2 km, respectively. A linear interpolation has been applied to SABER data to make a comparison with SCIAMACHY data. This may underestimate the SABER data at peak altitudes and overestimate the SABER data at two wings besides the peak altitudes.

Figure 5 shows the global spatial distributions of SABER OH 2.0 μm VERs (bottom) and the corresponding simulations (top) using SCIAMACHY data for the year 2007. A strong annual variation with a maximum in April and a semi-annual oscillation are visible in the radiance data over the equator region, as it was also found by Teiser and von Savigny (2017) in a study of SCIAMACHY OH(3-1) and OH(6-2) volume emission rates. It is obvious that SABER VERs are significantly larger than corresponding simulated values based on SCIAMACHY observations, as already stated. Comparing the SABER OH 1.6 μm VERs and the corresponding simulations leads to the same conclusion (not shown).

Figure 6 shows two pairs of scatter plots which elucidate the consistency of SABER "unfiltered" 1.6 μm (left) and 2.0 μm (right) VERs and corresponding simulated values based on SCIAMACHY observations. Again, SABER data are systematically larger than the SCIAMACHY simulations. The SABER 1.6 μm channel data (left column in Figure 6) are 44% larger for the "unfiltered" data and 23% larger for the in-band data, if all altitudes and latitudes are considered simultaneously in one fit. For the 2.0 μm data (right column in Figure 6), the differences are 23% and 35% on average, respectively.

To illustrate whether this difference changes on long time scales, figure 7 shows the ratio of SABER "unfiltered" and in-band data to the corresponding simulations based on SCIAMACHY data from 2003 to 2011. For the OH 1.6 μm "unfiltered" (in-band) data, the ratio value varies roughly between 1.2 (1.0) and 1.3 (1.2) for 2003-2009, reaching 1.1 for 2010 and 1.36 for 2011. The ratio varies between 1.0 (1.1) and 1.1 (1.2) for the OH 2.0 μm "unfiltered" (in-band) data. The data indicate that

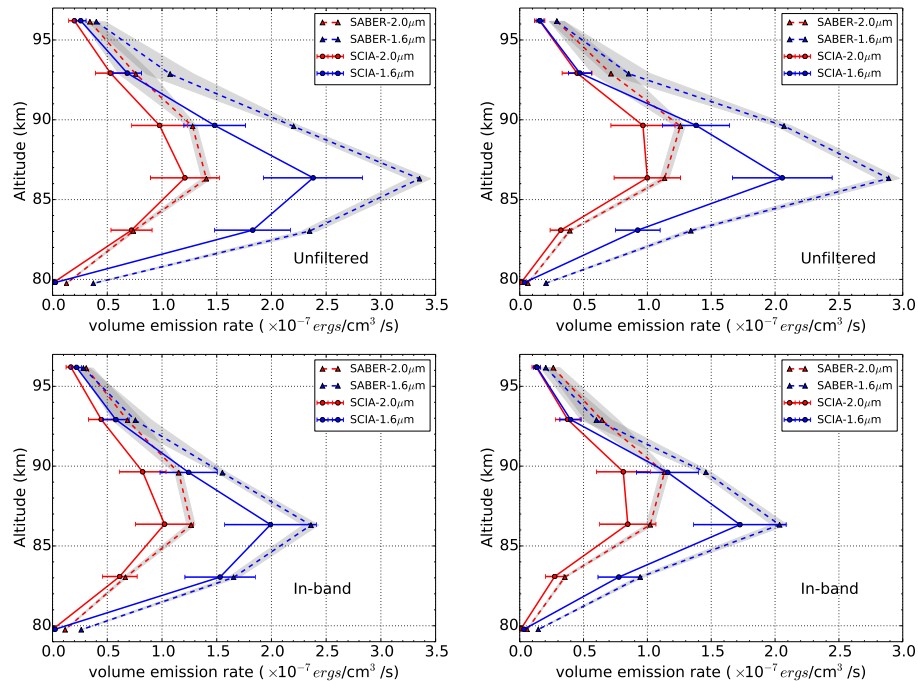

**Figure 4.** SABER 1.6 µm and 2.0 µm "unfiltered" (top) and in-band (bottom) volume emission rates and corresponding simulations from SCIAMACHY data for September 2005 at latitude bins 0°-20°N (left) and 20°N-40°N (right). The horizontal lines represent error bars considering the total uncertainties discussed in subsection 4.1. The grey shaded area represents the observed accuracy of SABER 1.6 µm and 2.0 µm channels.

there are no significant variations in the slope of SABER data versus SCIAMACHY simulations from 2003 to 2011 and that there is a systematic bias between them in general.

## 5   Conclusions

Near-infrared OH nightglow emissions measured by SCIAMACHY channel 6 were used in this study to simulate SABER 1.6 µm and 2.0 µm radiance measurements to assess systematic differences between the two measurements. Two different SABER data products are used for this comparison: So called in-band data, which are the data directly obtained from the measurements and "unfiltered" data. For the latter, the shape, width, and transmission of the instrument's broadband filters has been considered, and the fraction of OH lines passing the interference filter has been "upscaled" to obtain total band intensities of the corresponding vibrational transitions (Xu et al., 2012). If, however, in-band data is used, the data user has to apply the broadband filter transmission curve to the model data himself. This procedure has decisive advantages, because no a-priori assumptions have to be made to upscale partial measurements of OH vibrational bands to total band intensities. This allows to use consistent datasets of Einstein coefficients in all processing steps.

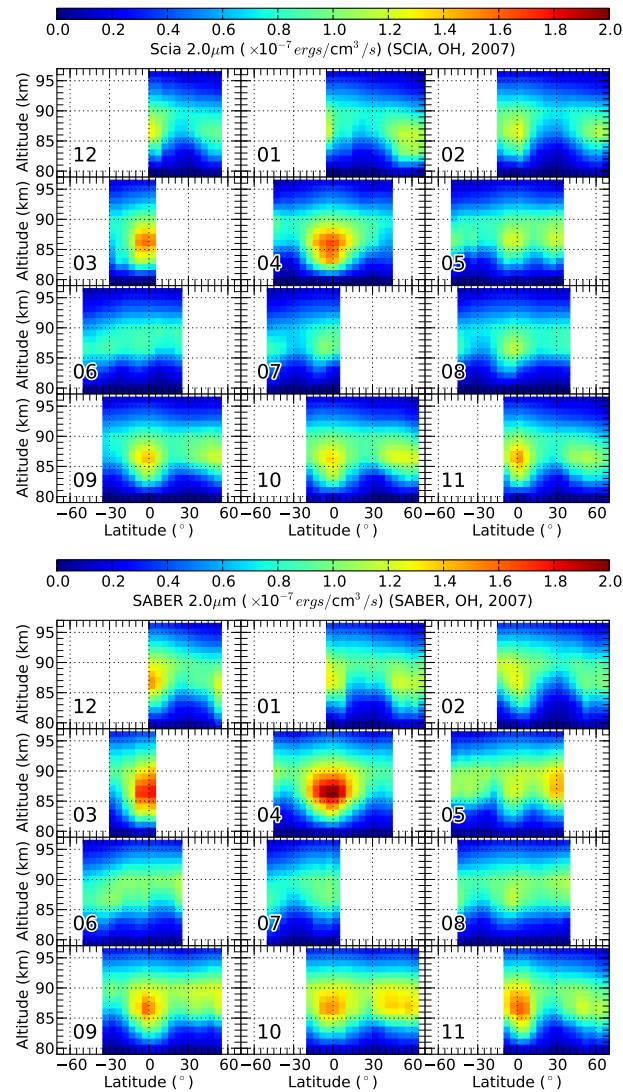

**Figure 5.** Latitude-altitude cross sections of monthly zonal median SABER "unfiltered" 2.0 μm volume emission rates (bottom) and corresponding simulations (top) from SCIAMACHY data for the year 2007. The numbers represent the month of the year.

When SABER OH in-band data are compared to model simulations using SCIAMACHY data, the typical differences are 35% for 2.0 μm and 23% for 1.6 μm radiances, whereas the differences are 23% and 44% for the "unfiltered" data, respectively. The significance or uncertainty of these differences is affected by uncertainties in the Einstein coefficients used to "map" SCIAMACHY to SABER data. For the in-band and "unfiltered" data, this uncertainty is estimated to be about 20% for the OH 1.6 μm channel and 26% for the OH 2.0 μm channel. Considering the radiometric uncertainty of both instruments, which is

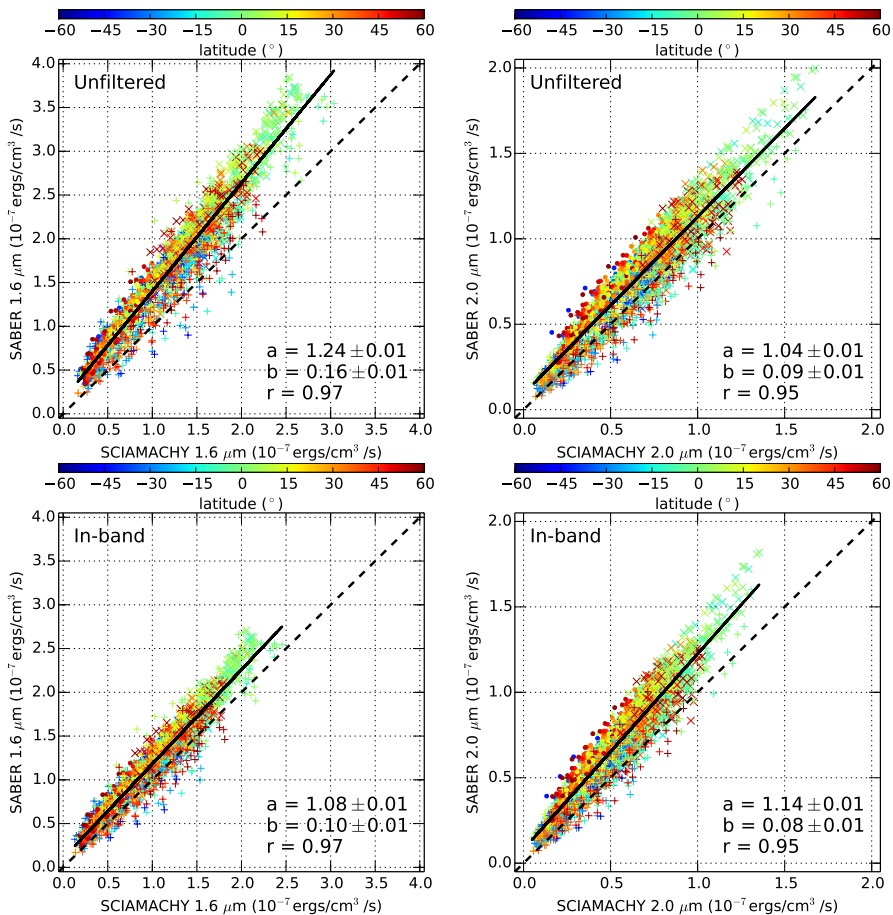

**Figure 6.** Scatter plots of SABER 1.6 μm (left) and 2.0 μm (right) volume emission rates versus the corresponding simulations using SCIAMACHY data for the year 2007. The color bar shows the latitude. The plus marker indicates the data at 80-85 km, the x marker represents the data at 86-90 km, and the point marker shows the data at 91-95 km. The solid line shows the linear fit to the data. a and b represent the slope and y-intercept of the fitting line, respectively. r represents the correlation coefficient of the fitting.

estimated to be about 1% for SCIAMACHY and 3-20% for SABER, OH 2.0 μm in-band and "unfiltered" data agree within their combined uncertainties; OH 1.6 μm in-band data also agree remarkably well, but not for the "unfiltered" 1.6 μm data.

     The OH 2.0 μm data measured by SABER and O($^1$S) green line emission and OH(9-6) nightglow observed by SCIAMACHY were used in the past to obtain atomic oxygen abundances. Significant differences in atomic oxygen absolute values were reported (Kaufmann et al., 2014; Mlynczak et al., 2018; Zhu and Kaufmann, 2018). These differences are of similar magnitude

as uncertainties in the Einstein coefficients and other model parameters used in the retrieval of those data.

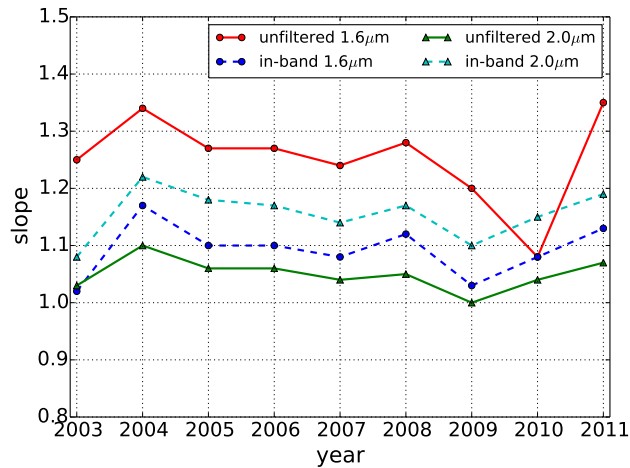

**Figure 7.** Slope of SABER 2.0 μm and 1.6 μm volume emission rates versus the corresponding simulations using SCIAMACHY data from 2003 to 2011.

*Data availability.* The SCIAMACHY Level 1b Version 8 data used in this study are available at ftp://scia-ftp-ds.eo.esa.int. SABER Version 2.0 data are available at http://saber.gats-inc.com. Derived OH volume emission rate data are available on request.

*Competing interests.* The authors declare that they have no conflict of interest.

*Acknowledgements.* Q. Chen, Q. Gong, J. Liu, and D. Wei were supported in their work by the China Scholarship Council. M. Kaufmann was supported by Forschungszentrum Jülich.

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
