# Peer review of "A comparison of OH nightglow volume emission rates as measured by SCIAMACHY and SABER"

_Atmospheric Measurement Techniques, 2019_

## Referee Comment (RC1) · Christian von Savigny (Referee) · 11 Oct 2019

General comments

This is a generally well written manuscript dealing with the comparison of OH nightglow emission rates obtained from SABER and SCIAMACHY limb measurements. These satellite measurements cannot be directly compared, because the spectral regions covered by the two instruments do not fully overlap. Instead, the spectrally resolved SCIAMACHY measurements are used to retrieve the concentration of OH in the relevant vibrational states. These concentration profiles are then used to forward model the OH emission rate profiles as measured by SABER. The results are of interest to the aeronomy/airglow community and I recommend publication of the paper subject to

(mainly) minor corrections. I ask the authors to consider the comments and suggestions for improvements as described below.

Specific comments

Line 5: "OH 1.6 $\mu$m and 2.0 $\mu$m radiances as measured by SABER were retrieved from OH limb measurements recorded by SCIAMACHY"

This sentence is somewhat misleading, particularly the "retrieved from". In your study, OH concentration profiles were retrieved from SCIAMCHY limb measurements and these concentrations were then used to "simulate" SABER measurements, right? I suggest rephrasing the sentence – right now the sentence also suggests that SCIAMACHY measurements around 2.0 micron were used, which is not the case.

Line 7: "Systematic deviations of up to 88% were found"

I my opinion the abstract is too negative and not representative of the obtained results. Only the large differences are mentioned. However, the mean differences are on the order of 10% for the 2.0 micron channel and 35% for the 1.6 micron channel. I suggest mentioning this as well.

Line 46: "absolute volume radiances"

Do you mean volume emission rates? To my knowledge "volume radiance" is not standard terminology. Radiance (usually) has the units: photons / s / m2 / sr (and / nm in case of "spectral radiance").

Figure 1, y-axis label: is this really a "radiance"? Are the units correct? Radiance should also include a solid angle dependence, right? I assume this should be "volume emission rate"?

I also suggest mentioning in the figure caption, whether these are modelled or measured spectra.

Line 59: "In this study, only the spectral range of channel 6 up to 1650 nm"

I think you only used wavelenghts up to about 1600 nm. Channel 6+ (having a different detector material) starts shortly below 1600 nm. This would also be consistent with the shading in Figure 1.

Line 86: "monthly zonal median data"

The median was probably determined for each altitude separately? Was there a specific reason to use the median rather than the mean?

Line 96: "by dividing the corresponding Einstein" -> "by dividing BY the corresponding Einstein"

Line 102: "Boltzmann factor" -> "Boltzmann constant" ?

Line 107: "the SCIAMACHY OH limb measurements can be expressed as"

The SCIAMACHY OH limb measurements can also be expressed in this form without the 2 conditions mentioned in the first part of the sentence. I suggest just stating that the SCIAMACHY measurements can be expressed in this form and that the two assumptions are made.

Line 109: "measured SCIAMACHY OH limb spectra measured."

Please delete one of the "measured"

Line 112: "of interested properties"

I suggest to replace this by "properties of interest". If the properties are interested in the retrievals, we don't know :)

Line 113: "In general, the inverse problem is ill-conditioned"

This is only minor point and I'm not asking for changes, but in your case, with the retrieval altitude grid being identical with the tangent height grid, the inverse problem is not ill-posed in the sense that there are more unknowns than knowns, right?

Line 114: "inverse issue" -> "inverse problem" ?

Figure 3: Perhaps colors can be used to highlight the 2 and 1.6 micron profiles? The symbols are quite small and difficult to identify.

Related to Figure 3: I think it is also worthwhile to show some sample OH(v) concentration profiles. They are the intermediate data product linking the SCIAMACHY and SABER measurements and are, therefore, quit important for this study.

Line 159/160: I suggest mentioning explicitly what reference profile was used to determine the relative differences. In case of large differences, this choice of reference will be important.

Line 161: "It was also found the positive deviations of SABER"

Please rephase, something is wrong here.

Same sentence and Fig. 4: I suggest discussing the difference between in-band and unfilered SABER data in a few additional sentences. It took me a while to figure out what's shown in Figure 4.

Line 165: "A strong annual oscillation was found over the equator region in April"

"A strong annual variation .. in April" doesn't really make sense, does it. You mean an annual variation with a maximum in April, I guess? There is also a semi-annual component in your figures, as, e.g. also clearly seen in Teiser & v. Savigny, JASTP, 161, 28-42, 2017.

Another general comment on the comparison of SCIAMACHY and SABER profiles: The SCIAMACHY and SABER volume emission rate profiles have different vertical resolutions. SCIAMACHY has a vertical resolution of about 4 km, SABER rather 2 km. The potential effects of this difference should be discussed, too. I'm not asking for more simulations etc., but only a qualitative discussion of the expected effects on the comparisons and the agreement.

Figure 7: y-axix label and caption: "slop" -> "slope"

[Figure]

---

## Referee Comment (RC2) · Konstantinos Kalogerakis (Referee) · 27 Oct 2019

**Review of manuscript amt-2019-328**

**General Comments**

This manuscript reports a comparison of nightglow OH Meinel band emission observations obtained by the SCHIAMACHY and SABER instruments on the Envisat and TIMED satellites, respectively. Several attempts in the literature to retrieve O-atom density profiles from OH Meinel band emissions have yielded inconsistent results. This situation appears to have improved recently, but significant discrepancies persist without a clear explanation. These retrievals depend on the assumed model for OH collisional relaxation and the associated kinetics parameters, the OH radiative transition probabilities (Einstein coefficients A), as well as the calibration of the observing instruments. The manuscript attempts to shed some light into this problem by comparing coincident measurements from the two instruments, thus removing the role of the OH kinetics model for production and loss processes and examining the possible influence of the other inputs. Establishing techniques to assess data from different space-based instruments that cannot be directly compared—as is the case here—is an important consistency check to better understand previous work and improve future analysis efforts.

The approach of the manuscript is sound overall. It is suitable for publication in Atmospheric Measurement Techniques. Below I will raise some concerns for the authors to consider. Depending on their findings, some additional analysis checks may be helpful, but at least some additional discussion is needed.

**Specific Comments**

Abstract: When reading the abstract for the first time, some parts are rather confusing. The meaning of the first three sentences is not clear and one needs to read the manuscript further for the abstract to make more sense. Somehow the references to the words "as measured" do not help and the second sentence "on the retrieval model to perform an inversion of OH(v) number densities in order to simulate OH ro-vibrational emission radiances using a non-linear regularized global fit technique" only complicates things further. The abstract needs to be well understood by itself and without referring to the rest of the manuscript. It must be clear to the readers that there are two sets of coincident / co-located measurements by these space-based instruments and one set is used to simulate the other and compare with it.

It seems to me the abstract (and possibly the manuscript in general) understates the uncertainties of the Einstein coefficients and exaggerates the observed deviations between the SCHIAMACHY observations and SABER simulations. Additional comments will be discussed below.

Main Manuscript:

- I am somewhat concerned that the truncation of the "overpopulated higher rotational levels" of the SCHIAMACHY spectra introduces systematic errors in this analysis. Non-local thermodynamic equilibrium conditions imply that we are dealing with a distribution that is not a true Boltzmann distribution. This applies to all rotational levels, including the ones with low rotational quantum numbers. For the lowest OH vibrational levels, it appears this effect is less significant and may be neglected in many cases, but the deviations become increasingly important for the highest vibrational levels and can, for example, lead to significant errors in the determination of the rotational temperatures. There is a sentence mentioning the authors performed a check of the effect of non-local thermodynamic equilibrium conditions and found it to be approximately 2%. It would be helpful to reconsider these checks and whether additional assessment is needed, and also provide some brief information on what these checks entailed. Moreover, it seems the SABER transmission windows include additional lines from other bands, e.g., there are a few lines of the 7-5 band in the 2.0-micron window. By neglecting these lines another systematic error is introduced, once again effectively "underestimating" the SCHIAMACHY measurements. Verifying that the above effects do not introduce significant bias in the simulation procedure and explicitly stating it would strengthen the manuscript.

- I find it rather difficult to accept the notion that the role of Einstein coefficients introducing bias is not significant. The fact that two rather similar sets of Einstein A coefficients were used provides some idea as to what differences can be expected, but the possibility of significant absolute systematic errors introduced by the coefficients cannot be excluded. The conversion of SCHIAMACHY observations to SABER simulations essentially depends on the ratio of the respective Einstein coefficients for the vibrational levels of interest, e.g., the ratios A(9-6)/A(9-7) and A(8-5)/A(8-6) for the SABER 2.0-micron channel. The values of these ratios for the HITRAN coefficient set are ~13% larger than those of Brooke et al., but approximately 26% larger than the Einstein coefficient set of van der Loo and Groenenboom (2007, 2008). With older A coefficient sets, the discrepancies can be much larger, but it is quite reasonable to consider that these three most recent Einstein coefficient sets are the most appropriate choices available.

- Figure 1 displays the region of overlap between the two instruments (note: it is difficult to view the shaded area, please modify this figure to make it legible). Would it be meaningful to make another comparison by using the overlapping spectral region of the 4-2 band to simulate the SABER 1.6-micron channel? If the fraction of the 4-2 band that

is covered is substantial, then such a comparison would be direct to some extent and would rely less on Einstein coefficients. Maybe testing one or two examples would clarify whether there is anything meaningful to be learned and any additional effort is warranted.

**Technical Corrections**

- Line 22: "imaged" or "monitored" may be more appropriate verb choices rather than "captured"
- Lines 112, 113, 114: "inverse" or "inversion"?
- Lines 162-163: "It was also found the positive deviations of SABER in-band data from the simulated values, especially for OH 2.0 $\mu$m data." Something is missing in this phrase.
- Line 167: "than the corresponding" instead of "than corresponding
- Figure 7: "slope" instead of "slop" in the label for the y axis and "slope" instead of "slops" in the caption
- Line 211: reference pages are missing
- Line 244: "Astrophys. J." This is the only journal that appears as an abbreviation (the preferred AMT format).
- Line 271: "Astrophysics" instead of "Atrophysics" (but still needs abbreviated title).

---

## Referee Comment (RC3) · Anonymous Referee #3 · 31 Oct 2019

The manuscript presents a detailed comparison of SCIAMACHY and SABER OH radiances which, given the large discrepancy between the atomic oxygen differences derived from both datasets, is really very important to be addressed quantitatively.

The paper is very well structured, with a clear motivation and a very good introduction to the field. The results are very important, particularly in the light of the significant differences in the atomic oxygen retrieved from both datasets. It is also well written in general, although it, needs some editing (at some points sentences are partially repeated). I am also listing some suggested writing changes and typos below. I therefore recommend the paper for publication in ACP but, before, I would like the authors address a couple of important points:

1) Although the differences found between the instruments are important themselves,

the authors should expand the laconic sentence that "the differences may be explained by the radiometric calibration of both instruments". I believe both instruments have been well characterized and estimates of their errors been given. The authors should discuss if the differences found are or not within the calibration errors of the instruments. Could they attribute the differences to a given instrument? e.g. on the basis that the calibration errors are smaller for one instrument than for the other?

2) I would also like the authors comment on the expected impact of these differences on the retrieved atomic oxygen from both datasets. Can the current O differences discussed in the introduction be explained by these differences? Would they reconcile the O differences or, on the other hand, would they still support or even enlarge the O differences?

Minor comments and suggested minor changes and typos:

Line 8. Worth to state here already which one (SABER/SCIA) is larger/smaller.

Line 30. Just over 1+1/2 solar cycles. Worth mentioning it is still measuring.

Lines 44-48. Consider some merging. Some information is somehow duplicated.

Legend of Fig. 1. Include that they are "simulated radiances".

Line 56. Delete "between two adjacent tangent heights". It is redundant.

Line 57. from vibrational states 2 to 9 at -> from UPPER vibrational states in the range of 2 to 9 at ...

Line 58. Bring all that information (particularly the radiometric calibration) to the discussion on the reason of the differences in Sec. 4.2 below.

Line 62. Start new paragraph with "The SABER ..."

Lines 73-76. Consider remove or re-write and merge with the next paragraph the sentence "The SCIAMACHY channel ... for SABER".

[Figure]

Legend of Fig. 2. Explain the meaning of the "fit" and "raw" symbols in the legend, so the reader do not have to read the text for understanding it.

Line 80. Mention here that rotational non-LTE is considered, and that it will be assessed later in the paper.

Near the end of page 4. About the method used when comparing SCIA and SABER radiances. Although probably the effects are small, I think it is more consistent to apply the SABER filters to the computed SCIA spectra and compared directly to the measured SABER radiances. In the way it has been done, by comparing with the "unfiltered" SABER radiances, the authors rely on the method used in SABER for unfiltering the radiances, e.g., in an OH model, which might be different, from that used in the retrieved OH radiances from SCIA.

Line 86. monthly zonal MEAN?

Line 97. ... dividing BY ...

Line 97. State that this equation is valid under the assumption of rotational LTE. And that rotational NLTE is considered later. Should not $E_v$ be actually $E_{rotational}$ in Eq. 1?

Line 111, "b" not only includes the Einstein coefficients, but also the other factors accompanying $n_v$ in Eq. 1.

Line 114. issue -> problem? Full stop -> comma? Consider re-writing.

Line 126 median -> MEAN?

Line 128. I believe the estimated effects of the temperature errors include not only the effects of temperature on the Einstein coefficient but also on the rotational populations (see Eq. 1).

Line 132. Probably worth to be clarified that the Einstein coefficient errors enter not only through the retrieval of SCIA, as it is described in the paragraph above, but also

through the A of the transitions of the SABER measured bands (which was not described in the paragraphs above).

Lines 138-139. Consider rewriting this sentence (somehow redundant with previous one).

Line 165. "A strong annual oscillation..." In the radiances or in the differences?

Line 169. Delete "here".

Fig. 4. I would suggest to use different line styles for SCIA and SABER? and/or use larger symbols.

Lines 187-189. Please see my two major points above.

Fig. 6. Right panel. The error of "b" is 0.00. Please, revise it.

Fig. 7. Slop -> slope? Both, in the legend and in the y-axis label. Is there any reason for the sudden drop in the slope in 2010? Please comment on it.

---

## Author Comment (AC1) · 7 Feb 2020

**Reply to comments of Reviewer #1**

We thank the reviewer for carefully reading the manuscript and his constructive and helpful comments and suggestions. They helped us to improve the paper in several aspects. We considered them point by point as illustrated below. We like to remark that line numbers mentioned in the reviewers comments refer to the first submission of the paper. We re-run the retrieval code to simulate OH radiances as measured by SABER from SCIAMACHY spectra and corrected radiance contamination to the previous unfiltered 1.6 $\mu$m simulations from other band emission lines due to the selected wavelength range. This problem was found when we run the retrieval using Einstein coefficients obtained by van der Loo and Groenenboom [2007, 2008] as suggested by the reviewer #2. The abstract was rephrased to make it more clear. Following the reviewer #3, we also simulated the in-band data as measured by SABER without considering the filter transmission effect for comparison.

**Specific Comments**:

**Line 5**: *"OH 1.6 $\mu$m and 2.0 $\mu$m radiances as measured by SABER were retrieved from OH limb measurements recorded by SCIAMACHY"*
*This sentence is somewhat misleading, particularly the "retrieved from". In your study, OH concentration profiles were retrieved from SCIAMACHY limb measurements and these concentrations were then used to "simulate" SABER measurements, right? I suggest rephrasing the sentence - right now the sentence also suggests that SCIAMACHY measurements around 2.0 micron were used, which is not the case.*
**Reply**: Following the reviewers' suggestion, we rephrased the abstract to make it more clear and to be well understood. Please refer to the abstract for details.

**Line 7**: *"Systematic deviations of up to 88% were found"*
*In my opinion, the abstract is too negative and not representative of the obtained results. Only the large differences are mentioned. However, the mean differences are on the order of 10% for the 2.0 micron channel and 35% for the 1.6 micron channel. I suggest mentioning this as well.*
**Reply**: We agree with the reviewer's suggestion. The mean difference is calculated using the formula:

$$\sum^{N} \frac{Radiance_{saber} - Radiance_{scia}}{Radiance_{scia}}/N$$

The average differences are on the order of 20% for the 2.0 $\mu$m channel and 40% for the 1.6 $\mu$m channel. The detailed description is here:

**On average, SABER "unfiltered" data is on the order of 40% at 1.6 $\mu$m and 20% at 2.0 $\mu$m larger than the simulations using SCIA-MACHY data.**

**Line 46**: *"absolute volume radiances"*
*Do you mean volume emission rates? To my knowledge "volume radiance" is not standard terminology. Radiance (usually) has the units: photons/s/m$^2$/sr (and /nm in case of "spectral radiance")*
**Reply**: Thanks for the reviewer pointing out this issue. We changed "volume radiance" to "spectrally averaged radiances at 1.6 $\mu$m and 2.0 $\mu$m as measured by SABER".

**Figure 1**: *y-axis label: is this really a "radiance"? Are the units correct? Radiance should also include a solid angle dependence, right? I assume this should be "volume emission rate"?*
*I also suggest mentioning in the figure caption, whether these are modelled or measured spectra.*
**Reply**: Yes, the unit is "volume emission rate", not "radiance". The label has been changed to "VER". The spectra are modelled, which is also mentioned in the caption.

**Line 59**: *"In this study, only the spectral range of channel 6 up to 1650 nm"*
*I think you only used wavelengths up to about 1600 nm. Channel 6+ (having a different detector material) starts shortly below 1600 nm. This would also be consistent with the shading in Figure 1.*
**Reply**: We only use the spectral range of channel 6 up to about 1589 nm. This was corrected accordingly in the main text and in Figure 1.

**Line 86**: *"monthly zonal median data"*
*The median was probably determined for each altitude separately? Was there a specific reason to use the median rather than the mean?*
**Reply**: Yes, the median was determined separately for each altitude. There was no specific reason for using the median rather than the mean, because we tested the median and mean spectra before and found no big differences between them.

**Line 96**: *"by dividing the corresponding Einstein"$\rightarrow$ "by dividing BY the corresponding Einstein"*
**Reply**: Corrected.

**Line 102**: *"Boltzmann factor"$\rightarrow$"Boltzmann constant"?*
**Reply**: Corrected.

**Line 107**: *"the SCIAMACHY OH limb measurements can be expressed as"*
*The SCIAMACHY OH limb measurements can also be expressed in this form*

*without the 2 conditions mentioned in the first part of the sentence. I suggest just stating that the SCIAMACHY measurements can be expressed in this form and that the two assumptions are made.*
**Reply**: Following the reviewer's suggestion, the sentence was rephrased as below:

**The SCIAMACHY OH limb measurements can be expressed as**

$$\boldsymbol{y} = \boldsymbol{F}(\boldsymbol{x}, \boldsymbol{b}) + \boldsymbol{\epsilon}$$

**In our setup we assume that each atmospheric layer emits OH airglow homogeneously, and we set the retrieval grid to be identical to the tangent altitude grid of the averaged OH limb measurements.**

**Line 109**: *"measured SCIAMACHY OH limb spectra measured."*
*Please delete one of the "measured"*
**Reply**: Done.

**Line 112**: *"of interested properties"*
*I suggest to replace this by "properties of interest". If the properties are interested in the retrievals, we dont know :)*
**Reply**: Yes, we definitely agree :). It is corrected.

**Line 113**: *"In general, the inverse problem is ill-conditioned"*
*This is only minor point and Im not asking for changes, but in your case, with the retrieval altitude grid being identical with the tangent height grid, the inverse problem is not ill-posed in the sense that there are more unknowns than knowns, right?*
**Reply**: A well-posed problem should meet the three Hadamard criteria: 1. Having a solution; 2. Having a unique solution; 3. Having a solution that depends continuously on the parameters or input data. An ill-posed problem is the one which does not meet at least one of the criteria. In our case, we do the retrieval from the SCIAMACHY limb spectra and there are more knowns (sampling points) than the unknowns, but the solution is not unique. That is why we call it ill-conditioned.

**Line 114**: *"inverse issue"→"inverse problem"?*
**Reply**: Corrected.

**Figure 3**: *Perhaps colors can be used to highlight the 2 and 1.6 micron profiles? The symbols are quite small and difficult to identify.*
*Related to Figure 3: I think it is also worthwhile to show some sample OH(v) concentration profiles. They are the intermediate data product linking the SCIAMACHY and SABER measurements and are, therefore, quite important for this study.*

**Reply**: The colors are used to highlight the 2 and 1.6 micron profiles. As suggested by the reviewer, corresponding OH(v) concentration profiles are added in Figure 3.

A sentence is added in the main text:

**Corresponding OH number density profiles as derived using Einstein coefficients from the HITRAN database at vibrational states 9, 8, 5, and 4 are also given.**

A sentence is added in the caption of Figure 3:

**Corresponding retrieved OH number densities of vibrational states 9, 8, 5, and 4 from SCIAMACHY data using HITRAN database (right).**

**Line 159/160**: *I suggest mentioning explicitly what reference profile was used to determine the relative differences. In case of large differences, this choice of reference will be important.*

**Reply**: The reference data are simulations from SCIAMACHY data. The sentences are modified:

**For the "unfiltered" data, deviations of SABER OH 1.6 $\mu$m measurements with respect to the corresponding simulations increase with altitude from 30-45% at 83 km to 55-80% at 96 km, depending on latitudes. The difference of SABER OH 2.0 $\mu$m measurements with respect to the corresponding simulations is 16% at 86 km.**

**Line 161**: *"It was also found the positive deviations of SABER"*
*Please rephrase, something is wrong here.*
*Same sentence and Fig. 4: I suggest discussing the difference between inband and unfiltered SABER data in a few additional sentences. It took me a while to figure out whats shown in Figure 4.*

**Reply**: This sentence was deleted and Figure 4 has been updated by adding a comparison of SABER in-band data and corresponding simulations using SCIAMACHY data. The difference between SABER in-band and unfiltered data has been discussed in Line 84-89. To make it more clear, some sentences are added in the main text.

**The top two plots show a comparison of the "unfiltered" data and the bottom two figures show the in-band data.**

**Line 165**: *"A strong annual oscillation was found over the equator region in April"*
*"A strong annual variation ... in April" doesnt really make sense, does it. You mean an annual variation with a maximum in April, I guess? There is also a semi-annual component in your figures, as, e.g. also clearly seen in Teiser & v. Savigny, JASTP, 161, 28-42, 2017.*

**Reply**: Thanks for the reviewer pointing out this problem. The sentence was rephrased.

**A strong annual variation with a maximum in April and a semi-annual oscillation are visible in the radiance data over the equator region, as it was also found by Teiser and von Savigny [2017] in a study of SCIAMACHY OH(3-1) and OH(6-2) volume emission rates.**

**General comment**: *Another general comment on the comparison of SCIA-MACHY and SABER profiles: The SCIAMACHY and SABER volume emission rate profiles have different vertical resolutions. SCIAMACHY has a vertical resolution of about 4 km, SABER rather 2 km. The potential effects of this difference should be discussed, too. Im not asking for more simulations etc., but only a qualitative discussion of the expected effects on the comparisons and the agreement.*

**Reply**: As pointed out by the reviewer, SCIAMACHY has a vertical resolution about 3.3 km and the vertical resolution of SABER is around 2 km. A linear interpolation has been performed to the SABER data for the purpose of making a comparison with SCIAMACHY data. This means that we may underestimate the SABER data at peak altitudes and overestimate the data at two wings besides the peak altitudes. A discussion is given in the first paragraph of section 4.2:

**It should be noted that SCIAMACHY and SABER have a resolution of about 3.3 km and 2 km, respectively. A linear interpolation has been applied to SABER data to make a comparison with SCIA-MACHY data. This may underestimate the SABER data at peak altitudes and overestimate the SABER data at two wings besides the peak altitudes.**

**Figure 7**: *y-axis label and caption: "slop" → "slope"*
**Reply**: Corrected.

**References**

Georg Teiser and Christian von Savigny. Variability of OH(3-1) and OH(6-2) emission altitude and volume emission rate from 2003 to 2011. *J. Atmos. Sol.-Terr. Phys.*, 161:28 – 42, 2017. ISSN 1364-6826. doi: https://doi.org/10.1016/j.jastp.2017.04.010. URL http://www.sciencedirect.com/science/article/pii/S1364682616303364.

Mark P. J. van der Loo and Gerrit C. Groenenboom. Theoretical transition probabilities for the OH Meinel system. *J. Chem. Phys.*,

126(11):114314, 2007. doi: http://dx.doi.org/10.1063/1.2646859. URL
http://scitation.aip.org/content/aip/journal/jcp/126/11/10.1063/1.2646859.

Mark P. J. van der Loo and Gerrit C. Groenenboom. Erratum: The-
oretical transition probabilities for the OH Meinel system [J. Chem.
Phys. 126, 114314 (2007)]. *J. Chem. Phys.*, 128(15):159902, 2008. doi:
10.1063/1.2899016. URL https://doi.org/10.1063/1.2899016.

---

## Author Comment (AC2) · 7 Feb 2020

**Reply to comments of Reviewer #2**

We thank the reviewer for carefully reading the manuscript and his/ her constructive and helpful comments and suggestions. They helped us to improve the paper in several aspects. We considered them point by point as illustrated below. We like to remark that line numbers mentioned in the reviewers comments refer to the first submission of the paper. We re-run the retrieval code to simulate OH radiances as measured by SABER from SCIAMACHY spectra and corrected radiance contamination to the previous unfiltered 1.6 $\mu$m simulations from other band emission lines due to the selected wavelength range. This problem was found when we run the retrieval using Einstein coefficients obtained by van der Loo and Groenenboom [2007, 2008] as suggested by the reviewer #2. The abstract was rephrased to make it more clear. Following the reviewer #3, we also simulated the in-band data as measured by SABER without considering the filter transmission effect for comparison.

**Specific Comments**:

**Abstract**: *When reading the abstract for the first time, some parts are rather confusing. The meaning of the first three sentences is not clear and one needs to read the manuscript further for the abstract to make more sense. Somehow the references to the words "as measured" do not help and the second sentence "on the retrieval model to perform an inversion of OH(v) number densities in order to simulate OH ro-vibrational emission radiances using a non-linear regularized global fit technique" only complicates things further. The abstract needs to be well understood by itself and without referring to the rest of the manuscript. It must be clear to the readers that there are two sets of coincident / co-located measurements by these space-based instruments and one set is used to simulate the other and compare with it.*
*It seems to me the abstract (and possibly the manuscript in general) understates the uncertainties of the Einstein coefficients and exaggerates the observed deviations between the SCHIAMACHY observations and SABER simulations. Additional comments will be discussed below.*
**Reply**: Thank the reviewer for pointing out this issue. Most part of the abstract was rephrased to make it more clear to the readers. For details, please refer to the abstract. Following the reviewer's suggestion, Einstein coefficients calculated by van der Loo and Groenenboom [2007, 2008] were also considered in this work to estimate the uncertainties of the Einstein coefficients.

**Main Manuscript**:

**1**: *I am somewhat concerned that the truncation of the "overpopulated higher rotational levels" of the SCHIAMACHY spectra introduces systematic errors in this analysis. Non-local thermodynamic equilibrium conditions imply that we are dealing with a distribution that is not a true Boltzmann distribution. This applies to all rotational levels, including the ones with low rotational quantum numbers. For the lowest OH vibrational levels, it appears this effect is less significant and may be neglected in many cases, but the deviations become increasingly important for the highest vibrational levels and can, for example, lead to significant errors in the determination of the rotational temperatures. There is a sentence mentioning the authors performed a check of the effect of non-local thermodynamic equilibrium conditions and found it to be approximately 2%. It would be helpful to reconsider these checks and whether additional assessment is needed, and also provide some brief information on what these checks entailed. Moreover, it seems the SABER transmission windows include additional lines from other bands, e.g., there are a few lines of the 7-5 band in the 2.0-micron window. By neglecting these lines another systematic error is introduced, once again effectively "underestimating" the SCHIAMACHY measurements. Verifying that the above effects do not introduce significant bias in the simulation procedure and explicitly stating it would strengthen the manuscript.*

**Reply**: As said by the reviewer, non-LTE conditions affect all rotational levels deviating from a Boltzmann distribution with kinetic temperature. From the work of Oliva et al. [2015] and Kalogerakis et al. [2018], each vibrational state follows a Boltzmann distribution at low rotational levels, but with a different rotational temperature $T_{cold}$, which may differ from the kinetic temperature. Monthly zonal median OH airglow measurements from SCIAMACHY and monthly zonal mean temperature from SABER were used in the work. We can not use the SCIAMACHY measurements to investigate the non-LTE effect due to the low spectral resolution, as Oliva et al. [2015] and Kalogerakis et al. [2018] did based on cross-dispersed cryogenic spectrometer measurements in the spectral range of 0.97 $\mu$m to 2.4 $\mu$m. A theoretical study using the method of Oliva et al. [2015] has been performed. A weighted combination of two Boltzmann distribution equations with cold and hot OH rotational temperatures ($T_{cold}$ and $T_{hot}$) was used to predict the observed intensities of OH emission lines. $T_{cold}$ and $T_{hot}$ were taken from the figure 2 of Oliva et al. [2015], as well as the fractions of the cold and hot molecules. Firstly, the calculated OH(9-7) band radiance is less than 1% larger than the one only considering the cold molecules. Secondly, the band-pass of the SABER interference filters capture the hot and the cold fractions of the rotational distribution. Therefore, this redistribution of energy is averaged out. Last but not least, Xu et al. [2012] investigated the temperature

dependence of the band-averaged Einstein coefficient. They found that the OH(9-7) band Einstein coefficient only changes by approx. 0.35% when the temperature increases from 200 K to 250 K. Therefore, the Non-LTE effect on the band-averaged radiance is less important than for rotational temperature retrieval. A detailed description of the checks are added in the text.

**The low spectral resolution of SCIAMACHY spectra does not allow to estimate this effect from the measured data. Therefore, we performed model simulations using the same approach and parameter sets as Oliva et al. [2015] to quantify the effect of incomplete thermalization on the spectral ranges used in this study. We calculated OH 1.6 $\mu$m and 2.0 $\mu$m VERs by considering only the cold rotational temperature and then obtained them using cold and hot temperatures together as Oliva et al. [2015] and Kalogerakis et al. [2018] did. It was found that differences between them are less than 2% for both SABER channels.**

The OH 1.6 $\mu$m and 2.0 $\mu$m channels include additional lines from OH(3-1) and OH(7-5), respectively. We assume two ideal filters for these two channels with upper cut-off wavenumbers at 5150 cm$^{-1}$ for 2.0 $\mu$m channel and 6400 cm $^{-1}$ for 1.6 $\mu$m channel. OH(7-4) and OH(3-1) nightglow emissions have been observed by SCIAMACHY channel 6 and the same retrieval procedure described in the main text applied to them to derive OH(7-5) and OH(3-1) emissions. The contributions of OH(7-5) and OH(3-1) to the two channels are about 3% and 1%, respectively. A statement is added in the main text as below:

**The SABER 1.6 $\mu$m and 2.0 $\mu$m channels also observe emission lines from OH(3-1) and OH(7-5), respectively. We estimated their influence on spectrally integrated radiances by the derivation of the corresponding emissions using SCIAMACHY OH(3-1) and OH(7-4) nightglow measurements. These simulations show that the contributions of OH(7-5) and OH(3-1) to the two channels are about 3% and 1% on average, respectively.**

**2**: *I find it rather difficult to accept the notion that the role of Einstein coefficients introducing bias is not significant. The fact that two rather similar sets of Einstein A coefficients were used provides some idea as to what differences can be expected, but the possibility of significant absolute systematic errors introduced by the coefficients cannot be excluded. The conversion of SCHIAMACHY observations to SABER simulations essentially depends on the ratio of the respective Einstein coefficients for the vibrational levels of interest, e.g., the ratios A(9-6)/A(9-7) and A(8-5)/A(8-6) for the SABER 2.0-micron channel. The values of these ratios for the HITRAN coefficient set*

*are 13% larger than those of Brooke et al., but approximately 26% larger than the Einstein coefficient set of van der Loo and Groenenboom (2007, 2008). With older A coefficient sets, the discrepancies can be much larger, but it is quite reasonable to consider that these three most recent Einstein coefficient sets are the most appropriate choices available.*

**Reply**: Following the reviewer's suggestion, Einstein coefficients calculated by van der Loo and Groenenboom [2007, 2008] are also considered in the work. As said by the reviewer, the differences of the band-averaged Einstein coefficient ratios between HITRAN and those of van der Loo and Groenenboom (2007, 2008) are the largest by comparing the three Einstein coefficient datasets. The resultant differences of the retrieved VERs for 1.6-micron and 2.0-micron are 19% and 26%, respectively. The differences can potentially explain most of the deviations between SABER data and the simulations from SCIAMACHY data. The main text has been modified accordingly.

**3**: *Figure 1 displays the region of overlap between the two instruments (note: it is difficult to view the shaded area, please modify this figure to make it legible). Would it be meaningful to make another comparison by using the overlapping spectral region of the 4-2 band to simulate the SABER 1.6-micron channel? If the fraction of the 4-2 band that is covered is substantial, then such a comparison would be direct to some extent and would rely less on Einstein coefficients. Maybe testing one or two examples would clarify whether there is anything meaningful to be learned and any additional effort is warranted.*

**Reply**: As suggested by the reviewer, the shaded area in Figure 1 is highlighted to make it more legible. The overlapping spectral region by SCIA-MACHY and SABER only covers about half of OH(4-2) emissions. In addition, there are a few OH(3-1) lines in the bandpass of SABER $1.6\mu$m channel. In total, SCIAMACHY measurements cover only about 30% of the total radiances in the SABER $1.6\mu$m channel directly. Therefore, the use of Einstein coefficients to transfer SCIAMACHY to SABER measurements cannot be avoided.

**Technical Corrections**:

**Line 22**: *"imaged" or "monitored" may be more appropriate verb choices rather than "captured"*
**Reply**: "monitored" is used.

**Lines 112, 113, 114**: *"inverse" or "inversion"?*
**Reply**: It is "inversion". Corrected.

**Lines 162-163**: *"It was also found the positive deviations of SABER in-band data from the simulated values, especially for OH 2.0 $\mu$m data." Something*

*is missing in this phrase.*

**Reply**: This sentence is deleted because a more detailed description is given: **Surprisingly, for the in-band data, the differences for the 1.6 $\mu$m and 2.0 $\mu$m channels are significantly smaller at most altitudes. They vary in a range of 8-28% (21-50%) and 8-60% (28-100%) from 83 km to 96 km at 0°-20°N (20°N-40°N).**

**Line 167**: *"than the corresponding" instead of "than corresponding"*
**Reply**: Corrected.

**Figure 7**: *"slope" instead of "slop" in the label for the y axis and "slop" instead of "slops" in the caption*
**Reply**: Corrected.

**Line 211**: *reference pages are missing*
**Reply**: Added.

**Lines 244**: *"Astrophys. J." This is the only journal that appears as an abbreviation (the preferred AMT format).*
**Reply**: We changed every journal's name to their abbreviation.

**Line271** : *"Astrophysics" instead of "Atrophysics" (but still needs abbreviated title).*
**Reply**: The abbreviation of the journal is used.

**References**

D. J. Baker, B. K. Thurgood, W. K. Harrison, M. G. Mlynczak, and J. M. Russell. Equatorial enhancement of the nighttime OH mesospheric infrared airglow. *Phys. Scr.*, 75(5):615, 2007. URL `http://stacks.iop.org/1402-4896/75/i=5/a=004`.

K. S. Kalogerakis, D. Matsiev, P. C. Cosby, J. A. Dodd, S. Falcinelli, J. Hedin, A. A. Kutepov, S. Noll, P. A. Panka, C. Romanescu, and J. E. Thiebaud. New insights for mesospheric OH: multi-quantum vibrational relaxation as a driver for non-local thermodynamic equilibrium. *Ann. Geophys.*, 36(1):13–24, 2018. doi: 10.5194/angeo-36-13-2018. URL `https://www.ann-geophys.net/36/13/2018/`.

E. Oliva, L. Origlia, S. Scuderi, Benatti, S., Carleo, I., Lapenna, E., Mucciarelli, A., Baffa, C., Biliotti, V., Carbonaro, L., Falcini, G., Giani, E., Iuzzolino, M., Massi, F., Sanna, N., Sozzi, M., Tozzi, A., Ghedina, A., Ghinassi, F., Lodi, M., Harutyunyan, A., and Pedani, M. Lines and continuum sky emission in the near infrared: observational constraints

from deep high spectral resolution spectra with GIANO-TNG. *Astron. Astrophys.*, 581:A47, 2015. doi: 10.1051/0004-6361/201526291. URL `https://doi.org/10.1051/0004-6361/201526291`.

Mark P. J. van der Loo and Gerrit C. Groenenboom. Theoretical transition probabilities for the OH Meinel system. *J. Chem. Phys.*, 126(11):114314, 2007. doi: http://dx.doi.org/10.1063/1.2646859. URL `http://scitation.aip.org/content/aip/journal/jcp/126/11/10.1063/1.2646859`.

Mark P. J. van der Loo and Gerrit C. Groenenboom. Erratum: Theoretical transition probabilities for the OH Meinel system [J. Chem. Phys. 126, 114314 (2007)]. *J. Chem. Phys.*, 128(15):159902, 2008. doi: 10.1063/1.2899016. URL `https://doi.org/10.1063/1.2899016`.

Jiyao Xu, Hong Gao, Anne K. Smith, and Yajun Zhu. Using TIMED/SABER nightglow observations to investigate hydroxyl emission mechanisms in the mesopause region. *J. Geophys. Res.-Atmos.*, 117(D2):n/a–n/a, 2012. ISSN 2156-2202. doi: 10.1029/2011JD016342. URL `http://dx.doi.org/10.1029/2011JD016342`.

---

## Author Comment (AC3) · 7 Feb 2020

**Reply to comments of Reviewer #3**

We thank the reviewer for carefully reading the manuscript and his/ her constructive and helpful comments and suggestions. They helped us to improve the paper in several aspects. We considered them point by point as illustrated below. We like to remark that line numbers mentioned in the reviewers comments refer to the first submission of the paper. We re-run the retrieval code to simulate OH radiances as measured by SABER from SCIAMACHY spectra and corrected radiance contamination to the previous unfiltered 1.6 $\mu$m simulations from other band emission lines due to the selected wavelength range. This problem was found when we run the retrieval using Einstein coefficients obtained by van der Loo and Groenenboom [2007, 2008] as suggested by the reviewer #2. The abstract was rephrased to make it more clear. Following the reviewer #3, we also simulated the in-band data as measured by SABER without considering the filter transmission effect for comparison.

**General comments**:

**1)**: *Although the differences found between the instruments are important themselves, the authors should expand the laconic sentence that "the differences may be explained by the radiometric calibration of both instruments". I believe both instruments have been well characterized and estimates of their errors been given. The authors should discuss if the differences found are or not within the calibration errors of the instruments. Could they attribute the differences to a given instrument? e.g. on the basis that the calibration errors are smaller for one instrument than for the other?*

**Reply**: The observed accuracy of SABER 1.6 $\mu$m and 2.0 $\mu$m channels is about 3% at 80-90 km and about 20% at 90-100 km ("SABER Instrument Performance and Measurement Requirements" published on http://saber.gats-inc.com/overview.php). The propagated radiometric uncertainty of SCIAMACHY channel 6 is about 1.2% [Zoutman et al., 2000]. Both instruments have been well characterized. The differences of SABER data and the simulations from SCIAMACHY data are not within combined calibration errors of two instruments. However, we simulated OH radiances as measured by SABER using Einstein coefficients calculated by van der Loo and Groenenboom [2007, 2008] as recommended by the reviewer #2 and found that the uncertainty resulted from different datasets of Einstein coefficient could potentially explain the large differences of SABER data and the simulations from SCIAMACHY data.

**2)**: *I would also like the authors comment on the expected impact of these differences on the retrieved atomic oxygen from both datasets. Can the*

*current O differences discussed in the introduction be explained by these differences? Would they reconcile the O differences or, on the other hand, would they still support or even enlarge the O differences?*

**Reply**: We have analyzed the impact of these differences on the retrieved atomic oxygen from the two datasets and found these radiance differences almost reconcile the retrieved atomic oxygen differences, especially at low altitudes. A paragraph is added at the end of the text:

**The OH 2.0 $\mu$m data measured by SABER and O($^1$S) green line emission and OH(9-6) nightglow observed by SCIAMACHY were used in the past to obtain atomic oxygen abundances. Significant differences in atomic oxygen absolute values were reported [Kaufmann et al., 2014, Mlynczak et al., 2018, Zhu and Kaufmann, 2018]. These differences are of similar magnitude as uncertainties in the Einstein coefficients and other model parameters used in the retrieval of those data.**

**Minor comments and suggested minor changes and typos**:

**Line 8**: *Worth to state here already which one (SABER/SCIA) is larger/smaller.*
**Reply**: Stated.

**Line 30**: *Just over 1+1/2 solar cycles. Worth mentioning it is still measuring.*
**Reply**: A sentence is added:
**covering one and half solar cycles, and is still measuring**

**Lines 44-48**: *Consider some merging. Some information is somehow duplicated*
**Reply**: The duplicated sentence in line 44-45 was deleted.

**Legend of Fig. 1**: *Include that they are "simulated radiances".*
**Reply**: "Simulated" is included.

**Line 56**: *Delete "between two adjacent tangent heights". It is redundant.*
**Reply**: Deleted.

**Line 57**: *from vibrational states 2 to 9 at → from UPPER vibrational states in the range of 2 to 9 at ...*
**Reply**: Corrected.

**Line 58**: *Bring all that information (particularly the radiometric calibration) to the discussion on the reason of the differences in Sec. 4.2 below.*
**Reply**: Done.

**Line 62**: *Start new paragraph with "The SABER ..."*
**Reply**: Done.

**Lines 73-76**: *Consider remove or re-write and merge with the next paragraph the sentence "The SCIAMACHY channel ... for SABER".*
**Reply**: The sentence was merged with the next paragraph.

**Legend of Fig. 2**: *Explain the meaning of the "fit" and "raw" symbols in the legend, so the reader do not have to read the text for understanding it.*
**Reply**: An explanation is given in the caption of Figure 2.
**raw: the raw limb spectra measured by SCIAMACHY; fit: simulated limb spectra as measured by SCIAMACHY from retrieval results.**

**Line 80**: *Mention here that rotational non-LTE is considered, and that it will be assessed later in the paper.*
**Reply**: It is mentioned in the text.
**details are discussed later.**

**Near the end of page 4**: *About the method used when comparing SCIA and SABER radiances. Although probably the effects are small, I think it is more consistent to apply the SABER filters to the computed SCIA spectra and compared directly to the measured SABER radiances. In the way it has been done, by comparing with the "unfiltered" SABER radiances, the authors rely on the method used in SABER for unfiltering the radiances, e.g., in an OH model, which might be different, from that used in the retrieved OH radiances from SCIA.*
**Reply**: We agree with the reviewer for applying the SABER filters to the simulated radiance data from SCIAMACHY spectra. We applied bandpass filters of SABER 1.6 $\mu$m and 2.0 $\mu$m channels to the simulated OH radiances from SCIAMACHY measurements, including contributions of OH(7-5) and OH(3-1) emissions. Corresponding discussions are given in the text.

**Line 86**: *monthly zonal MEAN?*
**Reply**: We have tested monthly zonal median and mean spectra before and found no big differences between them.

**Line 97**: *... dividing BY ..*
**Reply**: Corrected.

**Line 97**: *State that this equation is valid under the assumption of rotational LTE. And that rotational NLTE is considered later. Should not $E_v$ be actually $E_{rotational}$ in Eq. 1?*
**Reply**: A statement is added as below:
**This formula is only valid under the rotational local thermodynamic equilibrium (LTE) condition and deviations are discussed later.**

Here, $E_v(i)$ represents the rotational energy of the upper rotational state of the $i$th line and is equivalent to the $E_{(v,J)}(i)$.

**Line 111**: *"b" not only includes the Einstein coefficients, but also the other factors accompanying $n_v$ in Eq. 1.*
**Reply**: Yes, i.e. $\rightarrow$ **e.g.**

**Line 114**: *issue $\rightarrow$ problem? Full stop $\rightarrow$ comma? Consider re-writing.*
**Reply**: re-written. **inverse issue $\rightarrow$ inversion problem**; Full stop $\rightarrow$ comma

**Line 126**: *median $\rightarrow$ MEAN?*
**Reply**: Yes, here is "mean".

**Line 128**: *I believe the estimated effects of the temperature errors include not only the effects of temperature on the Einstein coefficient but also on the rotational populations (see Eq. 1).*
**Reply**: Yes, we agree with the reviewer. According to Equation 1, the change of temperature will redistribute the rotational populations first, and then affects the band-average Einstein coefficient. The effects of temperature on the band-averaged Einstein coefficient and on the rotational populations are not independent with each other. Both of them have been considered in the estimated effects.

**Line 132**: *Probably worth to be clarified that the Einstein coefficient errors enter not only through the retrieval of SCIA, as it is described in the paragraph above, but also through the A of the transitions of the SABER measured bands (which was not described in the paragraphs above).*
**Reply**: The estimation of the Einstein coefficient errors in the text have considered these two aspects: one enters through the retrieval of SCIAMACHY data; another one enters through the simulation when using the Einstein coefficient of the SABER measured bands. A sentence is added in the text;
**The uncertainty of the Einstein coefficient affects simulated VERs in two ways: In the retrieval of vibrationally excited OH from SCIAMACHY data and in the simulation of the SABER measurements.**

**Lines 138-139**: *Consider rewriting this sentence (somehow redundant with previous one).*
**Reply**: The sentence is rephrased:
**Therefore, we used these results as a proxy to estimate related uncertainties of the Einstein coefficients.**

**Line 165**: *"A strong annual oscillation..." In the radiances or in the differences?*

**Reply**: It is the radiance and specified:
**A strong annual variation with a maximum in April and a semi-annual oscillation were found in the radiance data over the equator region, as it was also found by Teiser and von Savigny [2017] in a study of SCIAMACHY OH(3-1) and OH(6-2) volume emission rates.**

**Line 169**: *Delete "here".*
**Reply**: Deleted.

**Fig. 4**: *I would suggest to use different line styles for SCIA and SABER? and/or use larger symbols.*
**Reply**: Different line styles were used for SCIAMACHY and SABER data.

**Lines 187-189**: *Please see my two major points above.*
**Reply**: Considered.

**Fig. 6 Right panel**: *The error of "b" is 0.00. Please, revise it.*
**Reply**: Revised.

**Fig. 7**: *Slop → slope? Both, in the legend and in the y-axis label. Is there any reason for the sudden drop in the slope in 2010? Please comment on it.*
**Reply**: The word "slope" is correct. The sudden drop in the slope in 2010 results from very discrete fitting data points in 2010 for SABER 1.6 $\mu$m channel by comparing to other years.

**References**

M. Kaufmann, Y. Zhu, M. Ern, and M. Riese. Global distribution of atomic oxygen in the mesopause region as derived from SCIAMACHY O($^1$S) green line measurements. *Geophys. Res. Lett.*, 41(17):6274–6280, 2014. ISSN 1944-8007. doi: 10.1002/2014GL060574. URL http://dx.doi.org/10.1002/2014GL060574.

M. G. Mlynczak, Hunt L. A., J. M. Russell III, and B. T. Marshall. Updated SABER night atomic oxygen and implications for SABER ozone and atomic hydrogen. *Geophys. Res. Lett.*, 45(11):5735–5741, 2018.

Georg Teiser and Christian von Savigny. Variability of OH(3-1) and OH(6-2) emission altitude and volume emission rate from 2003 to 2011. *J. Atmos. Sol.-Terr. Phys.*, 161:28 – 42, 2017. ISSN 1364-6826. doi: https://doi.org/10.1016/j.jastp.2017.04.010. URL http://www.sciencedirect.com/science/article/pii/S1364682616303364.

Mark P. J. van der Loo and Gerrit C. Groenenboom. Theoretical transition probabilities for the OH Meinel system. *J. Chem. Phys.*, 126(11):114314, 2007. doi: http://dx.doi.org/10.1063/1.2646859. URL `http://scitation.aip.org/content/aip/journal/jcp/126/11/10.1063/1.2646859`.

Mark P. J. van der Loo and Gerrit C. Groenenboom. Erratum: Theoretical transition probabilities for the OH Meinel system [J. Chem. Phys. 126, 114314 (2007)]. *J. Chem. Phys.*, 128(15):159902, 2008. doi: 10.1063/1.2899016. URL `https://doi.org/10.1063/1.2899016`.

Yajun Zhu and Martin Kaufmann. Atomic oxygen abundance retrieved from SCIAMACHY hydroxyl nightglow measurements. *Geophys. Res. Lett.*, 45(17):9314–9322, 2018. doi: 10.1029/2018GL079259. URL `https://agupubs.onlinelibrary.wiley.com/doi/abs/10.1029/2018GL079259`.

Erik Zoutman, Giljam Derksen, Henk Bokhove, and Ralph Snel. Error budget for SCIAMACHY calibration. Technical report, TPD, 2000.